# Kaposi Sarcoma as a Possible Cutaneous Adverse Effect of ChAdOx1 nCov-19 Vaccine: A Case Report

**DOI:** 10.3390/vaccines12101168

**Published:** 2024-10-14

**Authors:** Yan-Han Li, Yu-Tzu Lin, Shu-Han Chuang, Hui-Ju Yang

**Affiliations:** 1Division of General Practice, Department of Medical Education, Changhua Christian Hospital, Changhua 500209, Taiwan; 185014@cch.org.tw (Y.-H.L.); b101106100@tmu.edu.tw (S.-H.C.); 2Department of Education, China Medical University Hospital, Taichung 404327, Taiwan; jenny1998tw@gmail.com; 3Department of Dermatology, Changhua Christian Hospital, Changhua 500209, Taiwan; 4Department of Post-Baccalaureate Medicine, College of Medicine, National Chung Hsing University, Taichung 402202, Taiwan

**Keywords:** Kaposi sarcoma, ChAdOx1 nCov-19 vaccine, AstraZeneca vaccine, SARS-CoV-2, vaccine adverse effect

## Abstract

The COVID-19 pandemic prompted the rapid development of vaccines, including the ChAdOx1 nCov-19 (AstraZeneca) vaccine. While effective, adverse effects have been reported, including cutaneous manifestations. Kaposi sarcoma (KS), a vascular tumor linked to Kaposi sarcoma herpesvirus/human herpesvirus 8 (HHV-8), has seen increased detection during the pandemic. This study reports a case of classic cutaneous KS in a 79-year-old male following the first dose of the ChAdOx1 nCov-19 vaccine, without prior SARS-CoV-2 infection. The patient developed multiple reddish-blue papules on his legs and feet, confirmed as KS through histopathology. Treatment included radiotherapy and sequential chemotherapy with Doxorubicin. The potential reactivation of latent HHV-8 by the vaccine is explored through mechanisms involving the SARS-CoV-2 spike protein and adenovirus vector, which may induce immune responses and inflammatory pathways. Although establishing a direct causal link remains challenging, the case highlights the need for vigilance regarding KS reactivation post-vaccination. Further large-scale studies are warranted to elucidate the relationship between COVID-19 vaccines and latent virus reactivation, ensuring comprehensive safety assessments and informed public health decisions.

## 1. Introduction

Among the devastating pandemic of coronavirus disease-19 (COVID-19), collaborative efforts have been undertaken to develop vaccines serving as a critical countermeasure in the battle against severe acute respiratory syndrome coronavirus 2 (SARS-CoV-2). The spike protein of SARS-CoV-2 binds to angiotensin-converting enzyme (ACE2) receptors on target cells during viral entry, so most COVID-19 candidate vaccines have been developed to induce anti-spike protein immune responses [1,2].

According to the WHO dashboard, 56% of the total population has been vaccinated with a complete primary series of a COVID-19 vaccine as of December 2023 [3]. Among them, ChAdOx1 nCov-19 (AstraZeneca) is one of the most administered vaccines. It is a replication-defective chimpanzee adenovirus-vectored vaccine expressing the full-length severe acute respiratory syndrome coronavirus 2 (SARS-CoV-2) spike glycoprotein gene. In various studies, ChAdOx1 has been linked to favorable effectiveness against the incidence rate, hospitalization, and mortality rate of COVID-19 in both the first and second doses across different populations and geographical areas [4,5]. However, some adverse effects were also observed, including cutaneous reactions. Notably, nearly 2000 cases of herpes viral infections were reported in the UK between January and June 2021 following AstraZeneca vaccination [6].

Kaposi sarcoma (KS) is a low-grade vascular tumor caused by human herpesvirus 8 (HHV-8) infection. HHV-8 is a human oncovirus responsible for around 35,000 cases of cancer and 16,000 deaths worldwide each year. (Recent trends in KS incidence and mortality show some fluctuation. The incidence was 41,799 cases in 2018, decreased to 34,270 in 2020, and slightly increased to 35,359 in 2022. Mortality attributed to KS was 19,902 deaths in 2018, decreased to 15,086 in 2020, and then rose to 15,911 in 2022) [7,8,9]. These lesions typically appear at mucocutaneous sites, initially involving the lower extremities, but they have the potential to affect other organs and anatomical locations. Cutaneous KS is characterized by the appearance of purplish, reddish-blue, or dark brown/black macules and papules on the skin that do not blanch under pressure and do not bleed upon palpation. These lesions are typically painless, non-pruritic, and vary in size (0.5 to 2 cm), being symmetrically distributed along the lines of skin tension. These lesions can remain unchanged for months to years, or they may grow rapidly within a few weeks and disseminate [10,11].

The detection of herpesviruses has become more common during the COVID-19 pandemic. Arman Shafiee et al. reported an increase in the detection of herpesviruses during the COVID-19 pandemic [12,13]. Among herpesviruses family, Humaira Lambarey et al. also noted elevated HHV-8 reactivation in non-hospitalized HIV-infected patients during the pandemic [14]. However, it remains unclear whether this cutaneous manifestation is due to the direct invasion of SARS-CoV-2, reactivation of latent HHV-8 viruses by the immunomodulatory capacity of anti-SARS-CoV-2 vaccines, or other factors. The potential reactivation of HHV-8 as a secondary reaction after vaccine protocols is an important consideration for diagnosis, as a few case reports have already indicated increased HHV-8 reactivation after vaccination without being infected [15,16].

Amidst the COVID-19 pandemic and the widespread administration of various vaccine protocols, this study aims to report a case of classic cutaneous KS that occurred following the first dose of the ChAdOx1 nCov-19 vaccine without being infected.

## 2. Case Report

The patient is a 79-year-old Taiwanese male with a history of controlled hypertension, hyperlipidemia, and coronary artery disease, for which he has been on long-term medications including Amlodipine, Valsartan, Clopidogrel, and Nebivolol. Notably, the patient has no history of any immunosuppressive medications or corticosteroid use. He received the first dose of the ChAdOx1 nCoV-19 vaccine in July 2021. He has no history of SARS-CoV-2 or HIV infection. Two days after vaccination, he developed skin rashes on all four limbs and sought help at the emergency room, where he was prescribed antihistamines, leading to the subsidence of symptoms. Four months later, he returned to the dermatology outpatient department with multiple reddish-blue papules on bilateral legs and feet (Figure 1A,B), along with two larger reddish erythematous papules on his left calf (Figure 1C,D). He denied any other discomforts and stated that he had not previously experienced similar lesions on his body. These lesions appeared gradually after vaccination and had enlarged recently. Excision of the two larger papules on the left calf was performed. The pathological report revealed Kaposi sarcoma, characterized by spindle-shaped tumor cells arranged in vascular spaces, positive for CD34, SMA, and HHV-8, negative for desmin (Figure 2).

Radiotherapy was administered within 2 months after diagnosis, with volumetric modulated arc therapy 4000 cGy/20 fractions to the right foot and left foot and lower leg, and electron beam therapy 4000 cGy/20 fractions to the left upper leg lesion. Three cycles of sequential chemotherapy with Doxorubicin at 20 mg/m^2^ were administered for locally advanced disease in the following 3 months, with no obvious systemic side effects noted. Following these treatments, no new recurrent lesions were found.

## 3. Discussion

The diagnosis of KS is typically suspected based on the appearance of characteristic lesions and their distribution on the skin. Histopathological confirmation of a diagnosis of KS remains the gold standard. KS characterized by a spindle cell proliferation of irregular, complex vascular channels dissecting through the dermis. Furthermore, immunohistochemistry has been used extensively in KS, with over 100 different primary antibodies evaluated, including CD31, CD34, D2-40, LYVE-1, and so on [10,11,17].

Kaposi sarcoma (KS) is classified into four distinct types based on clinical presentation and the distribution of lesions. These include classic, endemic African, HIV-related, and iatrogenic forms. Our patient, a male, exhibits characteristics most closely aligned with the classic form of KS. This type is predominantly found in patients over 50 years of age from Eastern European and Mediterranean backgrounds, features a male-to-female ratio of 17:1, and mainly affects the lower extremities [10,11,18]. To date, no antiviral treatment has been shown to be truly effective in eradicating latent HHV-8. However, several strategies have been utilized to manage classic KS, with the primary therapeutic goals being symptom palliation and reduction in lesion size. Local treatments such as surgical excision, intralesional chemotherapy, or local radiotherapy are commonly employed.

HHV-8 is essential for oncogenesis and is considered a direct carcinogen. However, its infection alone is insufficient for the development of KS; additional factors such as host immune dysfunction, local inflammation, or reactivation of latent virus are also necessary [11]. Several articles indicate that both COVID-19 infection and COVID-19 vaccination may lead to the incidence and/or reactivation of latent herpesviruses. For instance, Arman Shafee et al.’s meta-analysis highlights a potential association between COVID-19 vaccination and herpesvirus reactivation [12]. COVID-19 is linked to immunosuppression, especially in severe cases. Le Balc’h et al. noted that patients on prolonged mechanical ventilation often develop sepsis-associated immunosuppression, which increases the risk of reactivation of latent herpesviruses, such as herpes simplex virus (HSV) and cytomegalovirus (CMV) [19]. Paolucci et al. found a direct correlation between reduced CD8+ T cells and the presence of Epstein-Barr virus (EBV) DNA, suggesting that COVID-19 severity is associated with opportunistic viral reactivation [20]. Leoni et al. documented a patient with a history of Kaposi sarcoma who developed new skin manifestations during a COVID-19 infection, suggesting a correlation between the two conditions. They observed both HHV-8 and SARS-CoV-2 in the patient’s specimens, highlighting a potential interaction between these viruses [21]. It is essential to consider all viruses from the Herpesviridae family when managing patients who are either infected with or have recently been vaccinated against COVID-19.

In this instance, it seems that the ChAdOx1 nCov-19 vaccine served as one of the triggers required to induce KS. There have been several cases reported where physicians have postulated a connection between vaccinations, particularly COVID-19 vaccines, and the reactivation of latent KS. One such case involved a patient with a reactivation of human HHV-8 and the subsequent reappearance of KS lesions following a series of SARS-CoV-2 mRNA vaccinations (BNT162b2 and mRNA-1273) [22]. Other cases have reported KS development after COVID-19 vaccination [15,16]. Vaccines like MVA-BN have shown exacerbation of KS after administration [23]. Virological rebound of HHV-8 was also observed in classic KS patients after receiving seasonal influenza vaccination [24].

We postulate that the ChAdOx1 nCov-19 vaccination might induce latent HHV-8 reactivation, with multiple potential pathways reported to support this hypothesis.

One potential mechanism involves the spike protein of the ChAdOx1 nCov-19 vaccine. Research by Jungang Chen et al. involved transfecting the BCP-1 cell line, which harbors HHV-8, with vectors carrying the SARS-CoV-2 spike or nucleocapsid proteins, along with a control vector. Their findings revealed a significant elevation in the expression of key lytic genes, including RTA, ORF59, and ORF17, in cells transfected with either the spike or nucleocapsid proteins. Moreover, there was a noticeable increase in the production of mature virions, verified through qPCR, in cells that received these SARS-CoV-2 proteins [25]. Considering that the ChAdOx1 nCov-19 vaccine encodes the full-length SARS-CoV-2 spike glycoprotein, there is a conceivable risk that the vaccine could induce lytic reactivation of HHV-8 upon encountering cells infected with this virus.

Beyond the spike protein, the adenovirus vector in the ChAdOx1 nCov-19 vaccine may also trigger HHV-8 reactivation through immune response and inflammation. Research by Davor Nestić et al. revealed increased expression of IL-6, IL-8, IL-1β, and TNF-α in cells infected with a human adenovirus type 26-based vaccine vector, with IL-6 experiencing the most significant rise. This response is attributed to the Toll-like receptor (TLR) pathway, which promotes IL-6 production by activating the NF-kB transcription factors via MyD88 and TRIF [26]. Mark N. Polizzotto et al. have shown that HHV-8 flares correlate with rises in human IL-6 (hIL-6) or its viral homolog (vIL-6), triggering HHV-8’s lytic phase through the JAK/STAT3 pathway upon binding to its receptor. Such findings indicate that the adenovirus vector in the ChAdOx1 nCov-19 vaccine may facilitate IL-6 secretion, potentially causing HHV-8 lytic reactivation [27]. Additionally, the infection with this adenovirus vector could boost the expression of αvβ3 integrin, a molecule involved in angiogenesis and oncogenesis, further implicated in the development of KS [28].

Moreover, the ChAdOx1 nCoV-19 vaccine’s adenovirus vector comprises DNA strands capable of activating intracellular Toll-like receptors (TLRs) 3, 7, 8, and 9. As outlined by Gregory, Sean M., et al., HHV-8 reactivation may be prompted via the TLR 7/8 pathway through Interferon Regulatory Factors (IRFs). This mechanism not only initiates the lytic replication of HHV-8, allowing the virus to multiply, but also facilitates the virus’s escape from cells destined for apoptosis [29].

These potential mechanisms support our hypothesis that the vaccine could reactivate latent HHV-8. In our case, KS manifested several months post-vaccination with ChAdOx1 nCov-19, a period marked by the absence of other predisposing health issues or medical conditions. While establishing a direct causal link between the ChAdOx1 nCoV-19 vaccine and the onset of this neoplasm remains elusive, several contributing factors suggest this scenario is plausible. We conducted a survey of potential risk factors and found that, aside from the patient being older and male, he does not belong to a high-risk ethnicity. His immune status is not compromised, as he has no history of HIV or long-term use of immunosuppressive medications. Additionally, he does not have any other comorbidities such as chronic lymphatic obstruction, and there is no family history of KS or other cancers. Despite the unclear specifics of the underlying mechanism, the possibility of HHV-8 reactivation following vaccination protocols should not be overlooked.

However, these theories are currently speculative and should not discourage vaccination. Although Arman Shafee et al.‘s meta-analysis mentioned the possible association between COVID-19 vaccination and herpes virus reactivation, the focus was primarily on varicella-zoster virus (VZV), HSV, EBV, and CMV. Additionally, most of the vaccines included in that study were modified mRNA-based [12]. The relationship between HHV-8 reactivation and adenovirus-vectored vaccines, such as the ChAdOx1 nCoV-19 vaccine, requires further investigation. A thorough investigation is crucial to determine whether this reactivation could be a coincidence or a consequence, either related to reactivation by SARS-CoV-2 infection, the anti-SARS-CoV-2 vaccines, or other factors.

## 4. Conclusions

This case report suggests a possible connection between the ChAdOx1 nCoV-19 vaccine and the reactivation of latent HHV-8, leading to the development of KS. The patient, who had no prior history of KS or significant immunosuppressive conditions, developed KS several months after vaccination. Although no direct causal link can be firmly established, this case raises awareness about the potential for COVID-19 vaccines to trigger the reactivation of latent herpesviruses. The immune response induced by the vaccine, particularly the adenovirus vector, may play a role in this reactivation. Further research is needed to explore the mechanisms involved and to assess the risk of such adverse events in broader populations. These findings underscore the importance of monitoring patients with latent viral infections post-vaccination and conducting further studies to ensure the safety and efficacy of vaccination protocols.

## Figures and Tables

**Figure 1 vaccines-12-01168-f001:**
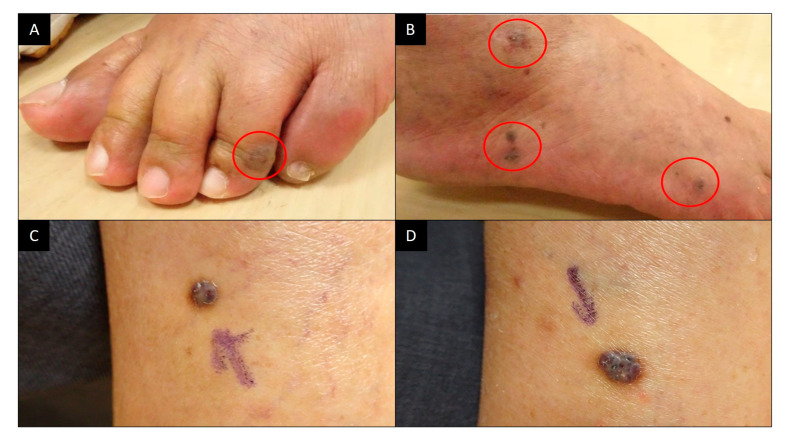
Clinical images of Kaposi sarcoma. (**A**) The patient presented with dark brown macules over the left foot (circled). (**B**) The patient presented with dark brown macules over the right foot (circled). (**C**,**D**) Two larger reddish erythematous papules on his left calf.

**Figure 2 vaccines-12-01168-f002:**
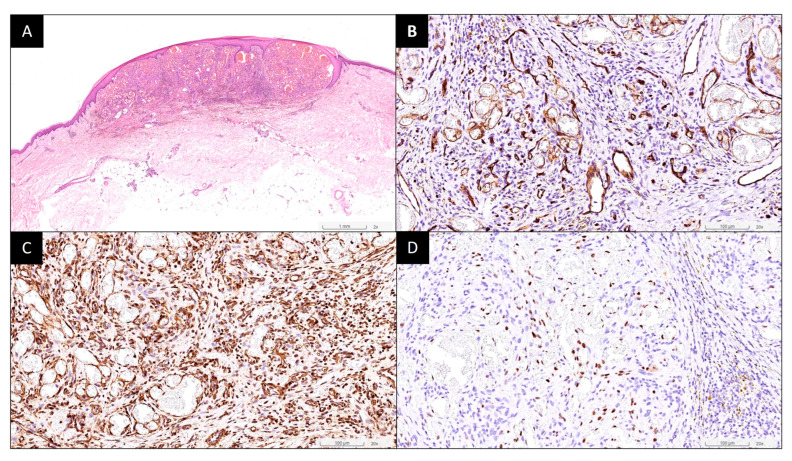
Representative histopathological images of tumor. (**A**) Neoplastic cells infiltrating the dermis and subcutaneous fat (H&E, ×40); strong and diffuse positive staining of neoplastic cells for CD31, ×200 (**B**), SMA, ×200 (**C**), and HHV-8, ×200 (**D**), for supporting the diagnosis of Kaposi sarcoma.

## Data Availability

All data underlying the results are available as part of the article and no additional source data are required.

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
