# Peer review of "Kaposi Sarcoma as a Possible Cutaneous Adverse Effect of ChAdOx1 nCov-19 Vaccine: A Case Report"

_vaccines, 2024, doi:10.3390/vaccines12101168_

Round 1
Reviewer 1 Report
Comments and Suggestions for Authors
Li et al describe a case of a 79-year old man with no prior history of either SARS-CoV-2 or HIV infection exhibiting gradual emergence of KS lesions months after receiving the ChAdOx1 nCoV-19 vaccine. The report is succinct with several relevant publications included on cases in the context of natural COVID infections. The report describes a single case where the appearance of KS lesions occurs after vaccination, and it is not clear if this is coincidental or causal (as has been noted by the authors).
Some additional details on the patient would be useful for readers - particularly medical practitioners:
-prior history of therapies for pre-existing hypertension, hyperlipidemia, coronary artery disease
-any immunosuppressive medications or corticosteroid uses
-additional risk factors for classic KS that can be ruled out
-any symptoms in addition to lesions appearing during the period following vaccination
The literature review should include additional references to studies on the association of COVID infections with incidence and/or reactivation of latent herpesviruses. Are there other published cases where physicians postulate a similar association between vaccination and KS?
Author Response
- Prior history of therapies for pre-existing hypertension, hyperlipidemia, coronary artery disease?
- Thank you for your insightful comments and suggestions. In response to your inquiry regarding the patient's prior history of therapies for pre-existing conditions, we have supplemented the Case section with details about his long-term medication regimen. Specifically, the patient has been treated with Amlodipine, Valsartan, Clopidogrel, and Nebivolol for his hypertension, hyperlipidemia, and coronary artery disease. If you have any further questions or require additional information, please let us know, and we will be happy to address them. Thank you once again for your valuable feedback.
- Any immunosuppressive medications or corticosteroid uses?
- Thank you for your insightful comments and suggestions. In response to your question about the use of any immunosuppressive medications or corticosteroids, we have clarified in the Case section that the patient has no history of any such medications. If you have any further questions or need additional details, please let us know, and we will be happy to assist. Thank you once again for your valuable feedback.
- Additional risk factors for classic KS that can be ruled out?
- Thank you for your insightful comments and suggestions. In response to your question regarding additional risk factors for classic Kaposi Sarcoma (KS) that can be ruled out, we have addressed this in the discussion section. We conducted a survey of potential risk factors and found that, aside from the patient being older and male, he does not belong to a high-risk ethnicity. His immune status is not compromised, as he has no history of HIV or long-term use of immunosuppressive medications. Additionally, he does not have any other comorbidities, such as chronic lymphatic obstruction, and there is no family history of KS or other cancers. If you have any further questions or require additional information, please let us know, and we would be happy to assist. Thank you once again for your valuable feedback.
- Any symptoms in addition to lesions appearing during the period following vaccination.
- Thank you for your insightful comments and suggestions. In response to your inquiry regarding any symptoms in addition to the lesions appearing after vaccination, we have provided further details in the Case section. The patient received the first dose of the ChAdOx1 nCoV-19 vaccine in July 2021 and developed skin rashes on all four limbs two days post-vaccination. He sought help at the emergency room, where he was prescribed antihistamines, which led to the subsidence of his symptoms. Additionally, when he returned to the dermatology outpatient department, he reported no other discomfort apart from the skin lesions. If you have any further questions or require additional information, please let us know, and we will be happy to assist. Thank you once again for your valuable feedback.
- The literature review should include additional references to studies on the association of COVID infections with incidence and/or reactivation of latent herpesviruses.
- Thank you for your insightful comments and suggestions. In response to your request for additional references regarding the association of COVID-19 infections with the incidence and/or reactivation of latent herpesviruses, we have included relevant studies in our discussion section. These references provide a comprehensive overview of the current understanding of how COVID-19 may contribute to the reactivation of latent herpesviruses. If you have any further questions or require additional information, please let us know, and we will be happy to assist. Thank you once again for your valuable feedback.
- Are there other published cases where physicians postulate a similar association between vaccination and KS?
- Thank you for your thoughtful comments and suggestions. In response to your inquiry about other published cases where physicians have postulated a similar association between vaccination and Kaposi Sarcoma (KS), we have addressed this topic in our discussion section. We have included relevant references that highlight cases and studies suggesting a potential link between vaccination and the development or reactivation of KS. If you have any further questions or need additional information, please let us know, and we would be happy to assist. Thank you once again for your valuable feedback.
Reviewer 2 Report
Comments and Suggestions for Authors
Thank you for inviting me to review the manuscript by Li et al. The authors reported a case of Kaposi Sarcoma several months after the ChAdOx1 nCoV-19 vaccination. The case is interesting, and the manuscript is well-written. I have the following comments:
Title: The title, “Kaposi Sarcoma as a Possible Cutaneous Adverse Effect of ChAdOx1 nCoV-19 Vaccine: A Case Report and Comprehensive Literature Review,” suggests a comprehensive review of the literature. However, the manuscript only references previously published systematic reviews by other researchers. Therefore, I recommend either removing “and Comprehensive Literature Review” from the title or conducting a more thorough review of the current evidence. This would require including a section following the case report that summarizes previous cases, along with a table containing information from previous case reports or studies.
Page 2, Lines 46-47: The authors state, “Notably, nearly 2,000 cases of herpes viral infections were reported in the UK between January and June 2021.[6]” It would be helpful to provide information on how many cases were reported from the same region during the same time period in previous years. Is there any evidence that this number is higher compared to the same population and time frame in prior years?
Page 2, Line 74: The sentence reads, “Months later, he returned to the dermatology outpatient department with multiple reddish-blue papules on bilateral legs and feet.” Please specify exactly how long after the vaccination the lesions appeared. Was it 2 months or 22 months? This detail is crucial for understanding the rate of lesion progression. Additionally, please clarify if the patient had noticed similar lesions before or if any family members have a history of cancer.
Author Response
Reviewer 2
- Title: The title, “Kaposi Sarcoma as a Possible Cutaneous Adverse Effect of ChAdOx1 nCoV-19 Vaccine: A Case Report and Comprehensive Literature Review,” suggests a comprehensive review of the literature. However, the manuscript only references previously published systematic reviews by other researchers. Therefore, I recommend either removing “and Comprehensive Literature Review” from the title or conducting a more thorough review of the current evidence. This would require including a section following the case report that summarizes previous cases, along with a table containing information from previous case reports or studies.
- Thank you for your insightful comments. In response to your suggestion, we have removed “and Comprehensive Literature Review” from the title. While we have referenced relevant literature within the manuscript, we acknowledge that we did not include a section summarizing previous cases or a table containing information from previous case reports or studies. If you have any further suggestions or questions, please let us know!.
- Page 2, Lines 46-47: The authors state, “Notably, nearly 2,000 cases of herpes viral infections were reported in the UK between January and June 2021.[6]” It would be helpful to provide information on how many cases were reported from the same region during the same time period in previous years. Is there any evidence that this number is higher compared to the same population and time frame in prior years?
- We apologize for any confusion in our previous expression. The statement, "Notably, nearly 2,000 cases of herpes viral infections were reported in the UK between January and June 2021," refers specifically to adverse reactions following vaccination. We have provided additional context in the manuscript regarding this point.
- Regarding herpes virus incidence in the UK, the statistics indicate a 40% reduction in genital herpes infection diagnoses between 2019 and 2020, occurring against a backdrop of stable rates since 2011. This decline likely reflects the decreased capacity of sexual health services to offer face-to-face consultations and changes in sexual behavior during the COVID-19 pandemic. While there are additional statistics for other types of herpes infections, such as ocular or oral herpes, they were not included in the manuscript as they do not directly relate to the main topic.(Reference: https://www.nice.org.uk/)
- Furthermore, looking at global trends for Kaposi Sarcoma (KS), recent data from the Global Cancer Statistics (GLOBOCAN) indicates fluctuations in incidence and mortality: there were 41,799 cases in 2018, decreasing to 34,270 in 2020, and slightly increasing to 35,359 in 2022. Mortality attributed to KS was 19,902 deaths in 2018, decreasing to 15,086 in 2020, and then rising to 15,911 in 2022. Thank you for your feedback, and please let us know if you have any further questions or concerns!
- Page 2, Line 74: The sentence reads, “Months later, he returned to the dermatology outpatient department with multiple reddish-blue papules on bilateral legs and feet.” Please specify exactly how long after the vaccination the lesions appeared. Was it 2 months or 22 months? This detail is crucial for understanding the rate of lesion progression. Additionally, please clarify if the patient had noticed similar lesions before or if any family members have a history of cancer.
- Thank you for your valuable feedback. We have made the following revisions in response to your comments: In the case section, we have changed "months later" to "four months later" to specify the exact duration after vaccination when the lesions appeared. Additionally, we have included the statement that he had not previously experienced similar lesions on his body. Furthermore, in the discussion section, we addressed the patient's risk factors and clarified that he does not have any family members with a history of cancer. If you have any further suggestions or questions, please let us know!
Reviewer 3 Report
Comments and Suggestions for Authors
Li et al highlight an important issue following widespread COVID-19 mandates that are being seen worldwide and are not limited to the AstraZeneca ChAdOx1 nCoV-19 vaccine. This is a well-written and well-documented case report. Apart from a few minor issues, this case report is ready for publication.
· L35 should be “angiotensin-converting enzyme 2 (ACE2)”
· L48 “linked” should be changed to “caused by”
· L49 should be “HHV-8 is a human oncovirus responsible for …”
· “HHV-8” should be used throughout the manuscript. Some instances use HHV8 (e.g. L80).
· L85 the “20 mg/m2” the “2” should be in superscript.
· Figures 1 and 2 should be larger and take up the whole width of the page and the frame letters (e.g. A, B, etc) do not need to be in brackets.
· Figure 1 – Frame A and B images should be swapped so the left foot is on the left side.
· “ChAdOx1 nCoV-19” should be two words but in some instances (e.g. L148, 156, 157) it appears as one.
· Please remove the sentence starting on L162 as this is an opinion and doesn’t relate to the rest of the paragraph.
· L169 “mRNA-based” should be “modified mRNA-based”.
Author Response
Reviewer 3
- L35 should be “angiotensin-converting enzyme 2 (ACE2)
- Thank you for your observation. We have corrected "L35" to "angiotensin-converting enzyme 2 (ACE2)" in the manuscript. If you have any further suggestions or questions, please let us know!
- L48 “linked” should be changed to “caused by”.
- Thank you for your feedback. We have made the correction on line 48, changing "linked" to "caused by." If you have any further suggestions or questions, please feel free to let us know!
- L49 should be “HHV-8 is a human oncovirus responsible for …”
- Thank you for your comment. We have updated line 49 to read, "HHV-8 is a human oncovirus responsible for..." If you have any additional suggestions or questions, please let us know!
- “HHV-8” should be used throughout the manuscript. Some instances use HHV8 (e.g. L80)
- Thank you for your suggestion. We have conducted a thorough check of the manuscript and ensured that "HHV-8" is used consistently throughout. Additionally, we have changed the two instances of "KSHV" to "HHV-8." If you have any further comments or questions, please feel free to reach out!
- L85 the “20 mg/m2” the “2” should be in superscript.
- Thank you for your observation. We have corrected line 85 to display "20 mg/m²" with the "2" in superscript. If you have any more suggestions or questions, please let us know!
- Figures 1 and 2 should be larger and take up the whole width of the page and the frame letters (e.g. A, B, etc) do not need to be in brackets.
- Thank you for your feedback regarding the figures. We will ensure that Figures 1 and 2 are resized to take up the entire width of the page. Additionally, we will remove the brackets around the frame letters (e.g., A, B, etc.). If you have any further suggestions or questions, please feel free to let us know!
- Figure 1 – Frame A and B images should be swapped so the left foot is on the left side.
- Thank you for your suggestion. We will swap the images in Frame A and B of Figure 1 so that the left foot is positioned on the left side. If you have any additional feedback or questions, please let us know!
- “ChAdOx1 nCoV-19” should be two words but in some instances (e.g. L148, 156, 157) it appears as one.
- Thank you for pointing that out. We will correct the instances of "ChAdOx1 nCoV-19" that appear as one word to ensure consistency throughout the manuscript. If you have any further suggestions or questions, please feel free to reach out!
- Please remove the sentence starting on L162 as this is an opinion and doesn’t relate to the rest of the paragraph.
- Thank you for your feedback. We will remove the sentence starting on line 162, as it is an opinion that does not relate to the rest of the paragraph. If you have any additional comments or questions, please let us know!
- L169 “mRNA-based” should be “modified mRNA-based”.
- Thank you for your suggestion. We will change "mRNA-based" to "modified mRNA-based" in line 169. If you have any further comments or questions, please feel free to reach out!
Round 2
Reviewer 2 Report
Comments and Suggestions for Authors
Thank you for revisions. I have no further comment.